# Investigating the Nutritional Properties, Chemical Composition (UPLC-HR-MS) and Safety (Ames Test) of *Atriplex halimus* L. Leaves and Their Potential Health Implications

**DOI:** 10.3390/plants14213350

**Published:** 2025-10-31

**Authors:** Maria Eleonora Foletti, Massimo Tacchini, Gianni Sacchetti, Annalisa Maietti, Mohamed Lamin Abdi Bellau, Marinella De Leo, Alessandra Guerrini

**Affiliations:** 1Pharmaceutical Biology Laboratory, Research Unit 7 of Terra & Acqua Tech Technopole Laboratory, Department of Life Sciences and Biotechnology, University of Ferrara, Piazzale Luciano Chiappini, 3, 44123 Ferrara, Italy; mariaeleonora.foletti@unife.it (M.E.F.); massimo.tacchini@unife.it (M.T.); gianni.sacchetti@unife.it (G.S.); 2Department of Chemistry, Pharmaceutical and Agricultural Sciences, University of Ferrara, Via Fossato di Mortara, 19, 44121 Ferrara, Italy; annalisa.maietti@unife.it; 3Sahrawi Refugee Camps, Sahrawi Ministry of Public Health, Tindouf 37000, Algeria; mohamedlamin.abdibellau@unife.it; 4Department of Pharmacy, University of Pisa, Via Bonanno 33, 56126 Pisa, Italy; marinella.deleo@unipi.it; 5Centre for Instruments Sharing of Pisa University (CISUP), University of Pisa, Lungarno Pacinotti 43/44, 56126 Pisa, Italy

**Keywords:** *Atriplex halimus* leaves, UPLC-HR-MS, polyphenol bioaccessibility, flavonoids, nutritional properties, Ames test

## Abstract

Motivated by the plant’s ethnopharmacological importance and the health conditions of the Sahrawi people, who have been living as refugees for over 50 years, this study comprehensively assessed the nutritional profile, secondary metabolite composition, in vitro bioaccessibility, and toxicological safety of *Atriplex halimus* L. leaves. The proximate analysis demonstrated richness in dietary fiber (44.41 ± 0.11 g/100 g) and essential macro/microelements, notably iron (142.0 ± 2.41 mg/100 g). The lipid profile features essential polyunsaturated fatty acids, specifically linoleic and α-linolenic acid, accounting for 40.6 ± 7.0% of total fatty acids. The UPLC-HR--MS characterization of two extracts tentatively identified 13 specialized metabolites, including uncommon flavonoids such as highly glycosylated forms of isorhamnetin and syringetin. Caffeic acid 3-sulfate and caffeic acid 4-sulfate were identified by NMR. Although in vitro antioxidant activity (DPPH/FRAP tests) was minimal, the traditional decoction showed high total polyphenol bioaccessibility (71.52 ± 0.46%) during simulated gastrointestinal digestion following the harmonized static protocol. The Ames test (using *Salmonella typhimurium* TA98 and TA1535) confirmed toxicological safety, as neither extract induced mutagenic or genotoxic effects. In conclusion, the robust nutritional composition, in vitro proven safety, and high polyphenol bioaccessibility suggest *A. halimus* leaves as a promising, nutrient-rich functional ingredient.

## 1. Introduction

*Atriplex halimus* L. (Amaranthaceae), a halophytic shrub, grows in arid and semi-arid regions of the Mediterranean Basin as far as Western Asia, including Portugal, France, Spain (and the Canary Islands), Italy, Greece, Turkey, Cyprus, Syria, Lebanon, Jordan, Tunisia, Morocco, Algeria, Egypt, and Saudi Arabia. It stands out for its adaptability to extreme environmental conditions and its unique properties. This species thrives in open, sunny areas with neutral to alkaline and often saline soils. It is also well-adapted to the temperature fluctuations and varying soil conditions of a Mediterranean climate, which has dry summers and wet winters. Its dense foliage effectively suppresses the growth of other herbaceous plants. Its ability to absorb heavy metals makes it worthwhile for phytoremediation. This same capacity to accumulate minerals explains why its leaves were traditionally eaten during times of famine. The leaves, characterized by a salty taste, are also eaten as salad and spices. Since ancient times, *A. halimus* has been a source of fodder for livestock. In traditional medicine, decoctions and infusions of *A. halimus* leaves are employed to treat cardiovascular diseases, gastrointestinal disorders, diabetes, and arthritis [1,2,3]. The Sahrawi people, who have been living as refugees in the hamada in the southwestern part of the province of Tindouf (Algeria) for almost 50 years, traditionally use this crude drug, known as *legtaf*, as food, a cooking spice, and a traditional health remedy [4]. It is relevant to note that the living conditions in the camps, marked by poor access to water, food, and health services, have resulted in severe health issues, including undernutrition, malnutrition, anemia, and diabetes, especially among children under five and pregnant women. Recent nutritional surveys (2019–2020) confirm a worrying trend: anemia rates sharply increased for pregnant women (from 60% to 78%) and children under five (from 28.4% to 50.1%) during that period. Moreover, severe malnutrition among children under five has increased from 4.7% in 2016 to 7.6% in 2019 [5].

Recent publications have provided limited advances in understanding the shrub’s secondary metabolite composition. It has been discovered that the leaves of *A. halimus* are rich in characteristic secondary metabolites, particularly flavonoids, such as quercetin, isorhamnetin and myricetin derivatives with unusual sugar chains [6,7,8].

Our research aims to highlight, through proximate and mineral analysis, that the traditional crude drug, *A. halimus* leaves, may have significant potential in the dietary supplementation of the Sahrawi population. Chemical characterization, using UPLC-HR-DAD-MS, aims to highlight specialized metabolites in decoction and hydroalcoholic ultrasound-assisted extract that may also be useful in treating diseases related to malnutrition. The in vitro antioxidant activity (DPPH and FRAP tests) may provide preliminary evidence of the ability to counteract oxidative stress associated with various disorders. In vitro digestion, conducted according to the harmonized static protocol described by Minekus et al. [9] (Cost Action INFOGEST), and drug safety tests (Ames test) complete the information picture for health-promoting [10]. This research design is preparatory for the subsequent in vivo evaluation of *A. halimus* leaves, as planned by the project that funded this study.

## 2. Results and Discussion

### 2.1. Proximate Analysis and Mineral Composition

The proximate analysis, carried out on the leaves of *Atriplex halimus* L. (Table 1), showed a humidity of 6.4 ± 0.16%.

The protein content was 11.1 ± 0.49 g/100 g, slightly lower than the value found by El-Amier et al. [11] (13.1 ± 1.1% on aerial parts) in Egypt and the value of total nitrogen determined by Meradi et al. [12] (17.7 ± 1.1% on leaves and stems), in Algeria. It has also been shown that this nutritional value varied significantly in the *A. halimus* leaves over the course of the seasons [13]. *A. halimus* showed a high content of fiber (44.41 ± 1.1%) and the percentage of insoluble fiber was found to be more concentrated than the soluble one (41.16 ± 1.1%). This is an interesting result because high-fiber diets were associated with the prevention and treatment of diseases like diabetes, which is also common in the Saharawi population [5]. Consequently, this crude drug could be a potential nutraceutical ingredient for preventing and treating this disease [14]. Lipids were detected in lower amounts (1.80 ± 0.05%) than the value published in the literature 4.29 ± 0.36% [11].

Concerning the fatty acid profile (Table 2), *A. halimus* leaves showed a typical composition of alophytic plants with a high percentage of palmtic acid (46.9%), linoleic acid (23.7%) and *α*-linoleic acid (16.8%). Although its overall lipid content is low, the presence of linoleic and *α*-linolenic acid is significant because they are essential fatty acids. *α*-linolenic acid, in particular, is an important anti-inflammatory agent that can help with various diseases [15].

The total ash content is more than three times higher than the data reported in the literature [10]. The analysis of minerals (Table 3) showed a very high concentration of Na, while the values of calcium and magnesium were comparable with data previously published [16] for *A. halimus* leaves. Calcium is a crucial macro-element for human health. It plays a fundamental role in muscle contraction, blood coagulation, and nerve impulse transmission. It also helps regulate cell permeability and the activity of numerous enzymes, such as those that promote insulin release. Finally, it is essential for the growth and strength of bones and teeth. The health benefits of potassium, detected in concentration of 1.56 ± 0.01 g/100 g, could be relevant for blood pressure, bone density and risk of kidney stones [17,18]. Of particular relevance are also the high concentrations detected for iron (142.0 ± 2.41 mg/100 g), manganese (11.07 ± 0.23 mg/100 g), and zinc (2.40 ± 0.16 mg/100). The high content of these minerals is beneficial for reducing health problems like growth delay and anemia, which are common among the Sahrawi people and are caused by dietary deficiencies [5].

### 2.2. Total Polyphenol Content and In Vitro Antioxidant Activity

The yield of the hydroalcoholic extract was 29.35 ± 0.89% and, for the decoction, it was 32.56 ± 1.25%.

Regarding the polyphenol content (TPC) in *A. halimus* leaf extracts, the value for the decoction (27.36 ± 0.82 mg gallic acid equivalent (GAE)/g of decoction) was lower than that reported by Bouaziz et al. [19] (37.93 ± 0.02 mg GAE/g), but higher than the concentrations reported by Ennoury et al. [20] (5.73 ± 1.40 mg GAE/g) (Table 4). It should be noted that the experimental extraction conditions of Ennoury et al. [20] involved the addition of 1M NaOH, which could markedly influence polyphenol recovery. Roubi et al. [21] also reported that the aqueous extract had a value of 337.61 mg GAE/g and the hydroalcoholic extract had a concentration of 569.94 ± 41.72 mg GAE/g. Even in the case of this latter result, our data were significantly lower (30.12 ± 1.01 mg GAE/g).

The difference between our results and the literature may depend on soil, environmental factors, harvest season, and manufacturing process, such as the reduction grade of crude drug, solvent/drug ratio, temperature, and time of the extraction method [22]. In our research, we replicated the preparation of a decoction and a hydroalcoholic extract as previously published by our research group [23,24], starting from shredded plant material, in accordance with the traditional formulation method of these preparations.

*A. halimus* extracts showed negligible antioxidant power. In fact, the DPPH assay exhibited much higher IC_50_ values compared to the Trolox (4.37 μg/mL): 0.827 mg/mL for the hydroalcoholic extract and 1.79 mg/mL for the decoction. Roubi et al. [21] showed in the DPPH assay for 70% hydroalcoholic extract an IC_50_ equal to 0.59 ± 0.12 mg/mL and a value of approximately 1.20 mg/mL for an aqueous extract of *A. halimus* leaves. Bouaziz et al. [19] described an IC_50_ of 0.950 mg/mL for the aqueous extract of *A. halimus*. Overall, the results in the literature did not provide significant values for antioxidant activity measured by the DPPH test and crucially, did not include a comparison of the IC_50_ value with a positive control.

The data obtained with the FRAP assay were also moderate (Table 4).

### 2.3. HPLC-MS-DAD Characterization

The HPLC-DAD qualitative analyses showed the same main compounds for both extracts.

The chemical composition was explored by means of UPLC-HR-MS analyses (Figure 1).

The identification of detected components was tentatively obtained by comparing HR full and fragmentation mass spectra with literature data (Table 5) and including molecules with mass error <5 ppm. A total of 13 compounds were annotated belonging to different classes of phenols, except for compound **4**, which as assigned as an aminoacid derivative. Among phenols, the simplest glycoside was compound **1**, assigned as a dihydroxybenzoic acid hexoside due to the detection of a product ion at *m*/*z* at 153.02 generated by the loss of a hexosyl unit (−162 u) in the MS/MS experiments [25]. Furthermore, several hydroxycinnamic acids linked to unusual moieties were attributed, including sulfated caffeic acid (compounds **2** and **3**), and both *p*-coumaric and ferulic acids conjugated with isocitric acid (compounds **5**, **7** and **9**). The sulfated phenolic acids were reported in marine organisms, rarely in plants, but a recent study highlighted their occurrence also in plant kingdom including edible species [26]. Unambiguous information about their structure can be obtained by diagnostic product ions in the fragmentation mass spectra at *m*/*z* 96.96 and 79.96, corresponding to HSO_4_^−^ and SO_3_^−^, respectively. The isolation from hydroalcholic extract allowed them to be chemically characterized using ^1^H and ^13^C NMR as *trans*-caffeic acid 4-sulfate and *trans*-caffeic acid 3-sulfate, according to previous reported data [27,28] as the following: *trans*-caffeic acid 3-sulfate: ^1^H NMR (400 MHz, DMSO-d_6_) δ 6.17 (d, *J* = 15.9 Hz, H2), 6.77 (d, *J* = 8.4 Hz, H5′), 7.04 (d, *J* = 15.9 Hz, H3), 7.07 (dd, *J* = 8.4 and 2.0 Hz, H6′), 7.32 (d, *J* = 2.0 Hz, H2′). ^13^C NMR (100 MHz, DMSO-d_6_) δ 117.4 (C5′), 120.5 (C2′), 124.1 (C6′), 125.2 (C2), 126.6 (C1′), 136.6 (C3), 140.8 (C3′), 150.7 (C4′), 171.8 (C1), and *trans*-caffeic acid 4-sulfate: ^1^H NMR (400 MHz, DMSO-d_6_) δ 6.23 (d, *J* = 15.9 Hz, H2), 6.87 (dd, *J* = 8.4 and 2.1 Hz, H6′), 6.96 (d, *J* = 2.1 Hz, H2′), 7.05 (d, *J* = 15.9 Hz, H3), 7.13 (d, *J* = 8.4 Hz, H5′). ^13^CNMR (100 MHz, DMSO-d_6_) δ 115.3 (C2′), 118.5 (C6′), 122.7 (C5′), 127.8 (C2), 133.0 (C1′), 136.1 (C3), 141.1 (C4′), 149.3 (C3′), 170.5 (C1). Similarly, isocitric acid conjugates have been rarely documented in the literature, probably due to misrepresentation of MS data that are in part superimposable with those of quinic acid conjugates [29,30]. The use of HR-MS allows to overcome this limit, enabling unambiguously differentiation between the two classes of compounds. The detection in MS/MS experiments of diagnostic product ions at *m*/*z* 155.00 and 111.01 led to assign the presence of isocitric acid residue, confirmed by the molecular formula obtained by HR data. Based on these findings, compound **5** was annotated as *p*-coumaroylisocitric acid, while compounds **7** and **8** were annotated as two isomers of feruloylisocitric acid [31].

**Table 5 plants-14-03350-t005:** The UHPLC-HR-MS data of compounds tentatively identified in the hydroalcoholic extract of *Atriplex halimus* leaves.

Peak	Compound	*t*_R_ (min)	Formula	[M − H]^−^ (*m*/*z*)	MS/MS Ions *	Error (ppm)	[M + H]^+^ (*m*/*z*)	MS/MS Ions	Ref.
**1**	Dihydroxybenzoic acid hexoside	3.5	C_13_H_16_O_9_	315.0716	153.02, **109.03**	−1.904	-	-	[25]
**2**	*trans*-Caffeic acid 4-sulfate	3.9	C_9_H_8_O_7_S	258.9911	**179.03**, 135.04, 96.96, 79.96	−2.703	-	-	[26]
**3**	*trans*-Caffeic acid 3-sulfate	4.1	C_9_H_8_O_7_S	258.9911	**179.03**, 135.04, 96.96	−2.703	-	-	[26]
**4**	3-(*N*-sulfonylindolyl)-d-lactic acid	4.8	C_11_H_11_NO_6_S	284.0227	266.01, 222.02, 204.06, 142.07, 96.96, 79.96	−2.465	-	-	[32]
**5**	*p*-Coumaroylisocitric acid	9.3	C_15_H_14_O_9_	337.0558	191.01, 173.01, 155.00, **111.01**	−2.077	-	-	[31]
**6**	Isorhamnetin pentasaccharide	10.2	C_43_H_56_O_28_	1019.2866 ([M + Cl]^−^ 1055.2634)	887.24, **314.04**, 315.05, 299.02	−1.864	1021.2996	449.11, **317.06**	[33]
**7**	Feruloylisocitric acid I	10.4	C_16_H_16_O_10_	367.0661	191.02, 173.01, 155.00, **111.01**	−2.724	-	-	[31]
**8**	Atriplexoside B or Isorhamnetin tetrasaccharide I	10.5	C_38_H_48_O_24_	887.2452 ([M + Cl]^−^ 923.2219)	755.20, 315.05, **314.04**, 299.02	−1.2398	889.2576	449.11, **317.06**	[6,7]
**9**	Feruloylisocitric acid II	10.8	C_16_H_16_O_10_	367.0661	191.02, 173.01, 155.00, **111.01**	−2.724	-	-	[31]
**10**	Atriplexoside B or Isorhamnetin tetrasaccharide II	11.4	C_38_H_48_O_24_	887.2452 ([M + Cl]^−^ 923.2219)	755.20, **315.05**, 299.02	−1.2398	889.2576	449.11, **317.06**	[6,7]
**11**	Syringetin trisaccharide I	12.0	C_34_H_42_O_21_	785.2132 ([M + Cl]^−^ 821.1885)	653.17, 345.06, **315.05**, 300.03	−1.7830	787.2265	479.11, **347.07**, 317.06	[34]
**12**	Atriplexoside A or Isorhamnetin trisaccharide	12.1	C_33_H_40_O_20_	755.2028 ([M + Cl]^−^ 791.1796)	**315.05**, 314.04, 300.03, 299.02, 271.02	−1.589	757.2159	**317.06**	[6,8]
**13**	Syringetin trisaccharide II	12.6	C_34_H_42_O_21_	785.2132 ([M + Cl]^−^ 821.1882)	**345.06**, 330.04, 315.05	−1.7830	787.2265	**347.07**, 317.05	[34]

* Ion base peaks are shown in bold. The peak numbers correspond to those of Figure 1.

The last region of the LC-MS chromatogram was populated by several flavonoid glycosides. According to the literature, reporting the occurrence of flavonol glycosides in *A. halimus* [6,7,8], all components (compounds **6**, **8**, **10–13**) were annotated as isorhamnetin and syringetin glycosides, based on diagnostic base ion peaks at *m*/*z* 314.04 and 315.05, corresponding to the two aglycons, respectively. For all compounds the correct position of each sugar unit could not be established, thus the number of monosaccharides in the sugar chain was proposed. The full and fragmentation MS data for compounds **8** and **10** are consistent with those of atriplexoside B, a bidesmosidic flavonoid isolated by Kabbash and Shoeib [6] or of an isorhamnetin tetrasaccharide with a sugar chain composed by two pentoses, a deoxyhexose, and a hexose such as 3′-methoxyquercetin-7-*O*-β-d-fucopyranosyl-(1→3)-β-d-glucopyranosyl-3-*O*-β-xylopyranosyl-(1→4)-β-xylopyranoside [7]. Analogously, mass spectra of compound **12** showed product ions attributable to both atriplexoside A, a bidesmosidic flavonoid isolated by Kabbash and Shoeib [6] or an isorhamnetin trisaccharide such as isorhamnetin 3-*O*-β-d-apiofuranosyl-(1→2)-*O*-[α-l-rhamnopyranosyl-(1→6)]-β-d-glucopyranoside, isolated by Vitiello et al. [8]. Finally, the compound **4** showed a molecular deprotonated ion at *m*/*z* 284.0227 consistent with the molecular formula C_11_H_11_NO_6_S. Product ions at *m*/*z* 266.01 ([M − H_2_O]^−^), 222.02 ([M − H_2_O − CO_2_]^−^), 142.07 ([M − H_2_O − CO_2_ − SO_3_H]^−^) led to annotate this substance as 3-(*N*-sulfonylindolyl)-d-lactic acid, a tryptophan derivative containing a sulfate group, previously isolated from *Entada rheedei* Spreng [32].

### 2.4. In Vitro Digestion

Polyphenol bioaccessibility of *A. halimus* leaves was assessed using an in vitro gastrointestinal digestion model. The procedure simulates the human digestive tract, with physiologically accurate conditions like fluid composition, pH levels, and the time food spends in the three step of digestion: salivary, gastric and intestinal. This allowed to observe how the polyphenols’ accessibility changed throughout digestion [9,35]. The content of released phenolics steadily increased with each step. In particular, since the salivary phase, there has been a significant recovery of polyphenols. The highly acidic gastric environment further enhanced polyphenol extraction and, to a much lesser extent, the neutral intestinal environment (Table 6).

At the end of the intestinal phase the solid residue, which had therefore undergone the entire digestive process, was extracted by decoction to compare the content of polyphenol with that of the extract obtained from the crude drug. The concentration determined in the extraction, obtained from the residue of the digestive process, was 85 ± 6 mg GAE/g CD, approximately 14% of the value of the decoction prepared from the crude drug. This result suggests that more than 85% of polyphenols dissolved in the in vitro digestion solutions and can be absorbed during digestion. No previous studies on the in vitro digestion of *A. halimus* have been reported in the literature; therefore the results should be considered innovative.

### 2.5. Ames Test

To evaluate the mutagenic potential of the hydroalcoholic extract and the decoction from *A. halimus*, the Ames test was performed using two strains of *Salmonella typhimurium*, TA98 and TA1535. The assays were conducted in the absence and pres-ence of metabolic activation (S9 mix). Both extracts were tested at three concentrations (0.5, 2.5 and 5 mg/plate) and compared to DMSO as a negative control. The results are summarized in Table 7.

For both strains, with and without metabolic activation, neither extract induced a two-fold or higher increase in the number of revertant colonies when compared to the negative controls’ spontaneous reversion rate. The mean number of colonies per plate fell within the spontaneous reversion ranges extrapolated from the laboratory’s previous experiments, which is in line with the literature data [36,37]. Moreover, no dose-dependent increase in revertant colonies was observed for either strain under either metabolic condition. These results indicated that the hydroalcoholic extract and the decoction did not induce mutagenicity in terms of either frameshift or base substitution mutations, regardless of metabolic activation. Therefore, under the experimental conditions employed, the *A. halimus* leaves extracts did not exhibit genotoxic effects as assessed by the Ames assay.

## 3. Materials and Methods

### 3.1. Plant Material

The leaves of *Atriplex halimus* L. were collected from a wild population of plants at Bir Lehlu (coordinates: 26°20′58″ N 09°34′32″ W, Western Sahara). The sample authentication was performed by Dr. Mohamed Lamin Abdi Bellau and Prof. Alessandra Guerrini. Wild plant material was dried at room temperature for 15 days. The voucher specimen (code no. ATR.022.001) is stored in the Herbarium of the University of Ferrara (Italy). The present research is compliant with the Nagoya protocol.

### 3.2. Proximate Analysis

The methodology for the determination of moisture, proteins, total mineral and lipidic content, lipidic composition, and dietary fiber (total, soluble, and insoluble) was performed as published in our previous research [23].

### 3.3. Mineral Composition Determination

Na, K, Mg, Zn, Ca, Fe, Cu, and Mn were determined according to previously described methods [23].

### 3.4. Ultrasound-Assisted Extraction (UAE) and Decoction

For the UAE, an aliquot of 40 g of shredded crude drug was extracted with 400 mL of a 70% ethanol (Sigma Aldrich, Milan, Italy) solution to maintain a drug-solvent ratio of 1:10 (*w*/*v*). The drug-solvent mixture thus obtained was placed in an ultrasonic bath (Ultrasonik 104X, Ney Dental International, MEDWOW, Nicosia, Cyprus) for 30 min. The extract was centrifuged and filtered with a Buchner filter to isolate the supernatant. The alcoholic component was removed using an IKA RV 10 digital rotary evaporator (Werke GmbH & Co. KG, Staufen im Breisgau, Germany) while the residual water was removed by freezing at −80 °C and subsequent freeze-drying (LIO5P lyophilizer, 5Pascal s.r.l., Milan, Italy) [24].

For the decoction, 200 mL of distilled water was added to 4 g of the crude drug, previously shredded, to obtain a drug-solvent ratio of 1:50 *w*/*v*. The drug-water mixture was then placed on a hot plate under magnetic stirring, brought to 100 °C, and then kept at the boiling point for 5 min. After cooling, centrifugation was performed at 6000 rpm for 5 min, followed by filtration with a Buchner filter to isolate the supernatant. The aqueous extract was freeze-dried [23].

### 3.5. Total Phenolic Content (TPC) and Antioxidant In Vitro Test

The TPC in extracts obtained from *A. halimus* crude leaves and intestinal digested, was determined using Folin-Ciocâlteu reagent (Sigma Aldrich, Milan, Italy), according to a previously described method [38]. Each was performed in triplicate, according to a previously described method. All samples were prepared at an initial concentration of 10 mg/mL in ethanol and/or water. Briefly, 0.1 mL of each sample solution was mixed with 7.9 mL of water and 0.5 mL of Folin-Ciocâlteu reagent. After exactly 2 min, 1.5 mL of Na_2_CO_3_ (Sigma Aldrich, Milan, Italy) 20% was added and the resulting solution was incubated in the dark and at room temperature for 2 h. Absorbance of final solutions was detected at 765 nm with a Jasco V-750 Double Beam UV-Visible Spectrophotometer (Hachioji, Tokyo, Japan) and polyphenol concentration was extrapolated from a calibration curve constructed by analyzing standard solutions of gallic acid (Sigma Aldrich, Milan, Italy) at a concentration range between 0 and 5 µg/mL. The results of TPC in extracts are expressed as milligram gallic acid equivalents per 100 g of crude drug. The assay was also performed on simulated gastrointestinal fluids obtained from in vitro digestion, to assess the bioaccessibility of polyphenols, defined as the fraction of *A. halimus* leaves TPC released in GI fluids.

The DPPH assay was performed following the procedure ideated by Brand-Williams et al. in 1995 [39], with some modifications [40]. A 0.208 mM DPPH (Sigma Aldrich, Milan, Italy) solution was combined with decreasing concentrations of extracts (from 2.5 mg/mL to 39.06 μg/mL) or Trolox (from 20 to 0.31 μg/mL), used as the positive control in a 96-well microplate, in triplicate, and incubated in the dark at room temperature for 40 min. The absorbance was then measured in triplicate at 515 nm against a blank using a microplate reader (680XR, Bio Rad, Laboratories, Inc., Hercules, CA, USA). The percentage of DPPH inhibition was calculated using the following formula: IDPPH% = [1 − (A_1_/A_2_)] × 100, where A_1_ is the absorbance of the testing sample solution, and A_2_ is the absorbance of the DPPH solution without the sample. The results were expressed as the IC_50_ value, which is the concentration of the sample required to achieve 50% inhibition of the DPPH radical. All experiments were conducted in triplicate.

The FRAP test was performed according to the method proposed by Guihua et al. [41], with some modifications. Samples were tested at a range of concentrations between 5 and 1 mg/mL. A 0.1 M acetate buffer (pH 3.6), a 10 mM 2,4,6-tripyridyl-s-triazine (TPTZ) solution (Sigma Aldrich, Milan, Italy) in 40 mM hydrochloric acid (HCl) (Sigma Aldrich, Milan, Italy), and a 20 mM ferric chloride (FeCl_3_) (Sigma Aldrich, Milan, Italy) solution were freshly prepared. Acetic buffer, TPTZ, and FeCl_3_ solutions were mixed in the ratio 10:1:1 (*v*:*v*:*v*) to obtain the FRAP reagent solution (Sigma Aldrich, Milan, Italy). To 0.1 mL of extracts or solvent (blank) was added 1.9 mL of reagent solution and the mixtures were incubated for 7 min. The absorbance was measured in triplicate at 593 nm, using a UV-VIS ThermoSpectronic Helios-γ spectrophotometer (Waltham, MA, USA). Standard Trolox solutions between 40 and 8 μg/mL were used to obtain the calibration curves. The FRAP values were expressed as µmol equivalents of Trolox for each gram of product.

### 3.6. UPLC-DAD-MS and Isolation of Pure Compounds

The LC-MS equipment was composed of a Vanquish Flex UHPLC coupled to a diode array detector (DAD) and a HR-MS Q Exactive Plus Orbitrap-based FT-MS with an electrospray ionization (ESI) source (Thermo Fisher Scientific Inc., Bremen, Germany). The hydroalcoholic extract and decoction of *A. halimus* were dissolved in MeOH (1 mg/mL) and water, respectively, centrifuged at 3500 rpm for 5 min, filtered by Phenex™ Teflon^®^ (PTFE) filter membranes (0.45 µm pore size, 47 mm diameter) and 5 µL were injected into a C18 Kinetex^®^ Biphenyl column (100 × 2.1 mm, 2.6 µm particle size) provided by a security Guard^TM^ Ultra Cartridge (Phenomenex, Bologna, Italy). The flow rate was 0.5 mL/min with a splitting system of 1:1 to the MS and DAD detectors, and the column oven temperature was set at 35 °C. The mobile phase was composed of a mixture of 0.1% *v*/*v* formic acid (Sigma Aldrich, Milan, Italy) in water (solvent A) and 0.1% *v*/*v* formic acid in methanol (Sigma Aldrich, Milan, Italy) (solvent B), following a linear gradient increasing from 5 to 65% solvent B within 21 min. The mass spectra were registered in a scan range of *m*/*z* 200–1500 in both ESI negative and positive ion modes, in full (70,000 resolution, 220 ms maximum injection time) and data-dependent-MS/MS scan. The ionization parameters were the following: spray voltage 3400 V (+ESI) and 3200 (−ESI), capillary temperature 290 °C, sheath gas (N_2_) 24 arbitrary units, auxiliary gas (N_2_) 5 arbitrary units, higher energy C trap dissociation (HCD) 18 eV. UV chromatograms were registered in a range of 200–600 nm, using 254, 280 and 325 nm as preferential channels. Data were elaborated by Xcalibur 4.1 software (Thermo Fisher Scientific Inc., Bremen, Germany). The isolation of caffeic sulfate was performed in a silica gel chromatographic column (silica gel 60 220–440 mesh, particle size: 0.035–0.070 mm, Sigma-Aldrich, Milan, Italy). For elution of 2 g of dried hydroalcoholic extract the mobile phase included ethyl acetate (Sigma Aldrich, Milan, Italy), water, formic acid, and acetic acid (Sigma Aldrich, Milan, Italy) with the 10:1:0.5:0.5 ratio, respectively. The solvents of collected fractions were partially reduced with a rotary evaporator and then lyophilized. The fractions were purified with RP-HPLC using the linear gradient described above.

### 3.7. NMR Characterization

The ^1^H-NMR and ^13^C-NMR were recorded in DMSO-d_6_ (Sigma Aldrich, Milan, Italy) solution in 5 mm tubes, at room temperature, with a Varian Mercury Plus 400 (Palo Alto, CA, USA), operating at 400 (^1^H) and 100 MHz (^13^C), respectively.

### 3.8. In Vitro Digestion

A procedure taken from Minekus et al. [9] was followed to simulate the digestive process to analyze and compare the amounts of polyhenols present in the crude drug at the end of digestion. Stock solutions of electrolytes, Simulated Salivary Fluid (SSF), Simulated Gastric Fluid (SGF) and Simulated Intestinal Fluid (SIF), as well as enzymes, were prepare according to the protocol provided in the reference publication. Briefly, 1 g of shredded crude drug was put in a conical tube and minced for 10 s mixed in 3.5 mL of SSF, 0.5 mL of 1500 U/mL salivary α-amylase solution prepared in SSF (α-amylase from porcine pancreas, ≥700 U/mg, MP biochemicals, Solon, OH, USA), 25 µL of 0.3 M CaCl_2_, and 975 µL of water. Bolus formation was completed with shaking incubation at 37 °C for 2 min, which mimicked the conditions of chewing and swallowing. To simulate the gastric phase, 6.5 mL of SGF, 1.6 mL of 25000 U/mL porcine pepsin stock solution in SGF (pepsin from porcine gastric mucosa 3200 U/mg, Sigma Aldrich, Milan, Italy), 5 µL of 0.3 M CaCl_2_ and 1.585 mL of water were added to the tube containing the bolus. 1 M HCl was added to reduce the pH to 3 (0.41 mL).

This phase was completed by incubating at 37 °C and shaking for 2 h to simulate peristalsis and gastric emptying. Finally, during the intestinal phase, the chyme from the gastric step was mixed with 11 mL of SIF, 5 mL of an 800 U/mL pancreatin solution in SIF (pancreatin from porcine pancreas 100 U/mg, TCI Chemicals, Tokyo, Japan), 2.5 mL of bile (bile extract porcine B8631, Sigma Aldrich, Milan, Italy), 40 µL of 0.3 M CaCl_2_, and 1.5 mL of water. Adding base (1 M NaOH) may be required to neutralize the mixture to pH 7.0. To simulate in vivo activity, intestinal digestion took 2 h at 37 °C and shaking. The experiment was conducted in triplicate. A first sample underwent only the salivary phase, the second both the salivary and gastric phases, and the third was subjected to the complete digestive process to isolate, through centrifugation at 6000 rpm for 5 min, the supernatants at the end of each phase. At the end of the intestinal phase, the centrifugation residue was recovered and extracted by decoction following the same procedure applied to the crude drug to compare extracts. From the comparison it was possible to assess the effectiveness of the digestion and the ability of simulated gastrointestinal fluids to extract polyphenols. In particular, the bioaccessibility of polyphenols, defined as the fraction of an external dose released from their matrix in the gastrointestinal tract, was calculated as follows:Bioaccessibility % = (CF/CI) × 100
where CF is the amount of polyphenols present in the digesta (chyme) and CI is the initial amount of polyphenols present in *A. halimus* leaves [35].

### 3.9. Ames Test

The mutagenicity was evaluated with the bacterial reverse mutation test, introduced by Bruce Ames in 1975, following OECD’s Test Guideline 471, plate incorporation method. The assay was performed on two *Salmonella typhimurium* mutant strains TA98 and TA1535. These strains, which require histidine to grow, were obtained from Molecular Toxicology Inc. (Boone, NC, USA). Fresh overnight cultures of the strains, at a concentration of approximately 10^8^ UFC/mL, were tested with and without adding an exogenous metabolic activator (a 5% S9 mix). This S9 (Sigma Aldrich, Milan, Italy) mix, a lyophilized post-mitochondrial supernatant from Aroclor 1254-induced male rat liver, was stored at −80 °C and is commonly used to activate pro-mutagens. Three different concentrations of each sample were prepared, at 50, 25 and 5 mg/mL in DMSO (Sigma Aldrich, Milan, Italy). The procedure for preparing the plates was as follows: in a sterile tube, 0.1 mL of test solution, 0.5 mL of either phosphate buffer (for assays without metabolic activation) or S9 mix (for assays with metabolic activation) and 0.1 mL of bacterial culture were mixed with 2 mL of molten top agar. The top agar was composed of 0.6% agar, 0.6% NaCl (Sigma Aldrich, Milan, Italy), and a 0.5 mM L-histidine/D-biotin solution (Sigma Aldrich, Milan, Italy). The content of each tube was thoroughly mixed and poured over the solid surface of minimal glucose agar plates (1.5% agar in 2% Vogel-Bonner medium E with 2% glucose) (Moltox, Boone, NC, USA). Dimethyl sulfoxide (DMSO) served as the negative control (100 µL per plate). The 2-nitrofluorene (Sigma Aldrich, Milan, Italy) (2 μg/plate) was used as a positive control without metabolic activation for TA98 and sodium azide (2 µg/plate) was used for TA1535. For metabolic activation, 2-aminoanthracene (Sigma Aldrich, Milan, Italy) was used as a positive control for both strains, at 5 μg/plate for TA98 and 10 μg/mL for TA1535. After preparation, the plates were incubated at 37 °C for 72 h. The number of his+ revertant colonies was then counted using a Colony Counter 560 Suntex (Antibioticos, Piave, Italy). A sample is considered mutagenic if there is a concentration-related increase over the evaluated range, at least one concentration shows a reproducible and clear, at least two-fold, increase in the number of his+ revertant colonies compared to solvent control and outside the distribution of fluctuation ranges extrapolated from the literature and previous experiments [36,37]. All tests were performed in triplicate to ensure reliability.

## 4. Conclusions

The conventional understanding of *Atriplex halimus* L. has primarily focused on its use in phytoremediation and as livestock fodder. Our findings contribute to the growing body of research validating the traditional use of its leaves in human health and nutrition.

This study demonstrated that the crude drug possesses a significant nutritional profile. Specifically, the high content of dietary fiber (44.41 ± 0.11 g/100 g) could be relevant for the prevention or management of dysmetabolic conditions, such as diabetes. Furthermore, the remarkable concentration of essential microelements, particularly iron (142.0 ± 2.41 mg/100 g), along with substantial levels of calcium and potassium, supports its utility in addressing potential mineral deficiencies. The presence of essential fatty acids, including linoleic and α-linolenic acid (40.6 ± 7.0% of total PUFA), further enhances its nutritional value.

The UPLC-HR-MS analysis, together with the NMR spectroscopy, made it possible to identify *trans*-caffeic acid 4-sulfate and *trans*-caffeic acid 3-sulfate. The other eleven molecules were only tentatively identified in the decoction and the hydroalcoholic extract. The LC-MS profiling highlighted the presence of uncommon flavonoids, such as highly glycosylated isorhamnetin and syringetin. These compounds represent secondary metabolites whose potential biological activities are still largely unexplored.

Although the extracts exhibited negligible in vitro antioxidant activity, the static in vitro digestion model demonstrated a high total polyphenol bioaccessibility (71.52 ± 0.46%). This suggests that the identified secondary metabolites are readily released from the matrix and potentially accessible for absorption in the gastrointestinal tract.

Finally, the toxicological safety of the traditional preparations (decoction and hydroalcoholic extract) was confirmed by the Ames test (using *S. typhimurium* TA98 and TA1535), which indicated the absence of mutagenic or genotoxic effects under the experimental conditions tested. In conclusion, the findings confirm that *A. halimus* leaves are a safe, highly bioaccessible, and nutritionally dense.

These characteristics contribute to validate its traditional usage and position it as a promising candidate for development as a functional food ingredient and a subject for future pharmacological investigations. This research provides the necessary groundwork for the subsequent in vivo evaluation of *A. halimus* leaves, as outlined in the funding project.

## Figures and Tables

**Figure 1 plants-14-03350-f001:**
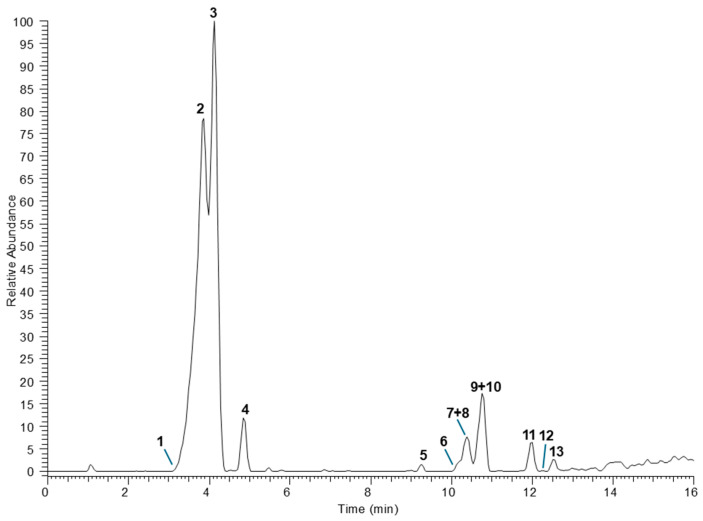
The UHPLC-HR-MS chromatogram of the hydroalcoholic extract of *Atriplex halimus* L. leaves registered in negative ESI mode. Peak numbers correspond to those listed in Table 5.

**Table 1 plants-14-03350-t001:** The proximate analysis of *Atriplex halimus* leaves.

Proximate Analysis	*A. halimus* Leaves Content
Humidity (g/100 g)	6.14 ± 0.16
Proteins (g/100 g)	11.20 ± 0.49
Lipids (g/100 g)	1.80 ± 0.05
Total ash (g/100 g)	36.20 ± 0.47
Total fiber (g/100 g)	44.41 ± 0.11
Insoluble fiber (g/100 g)	41.16 ± 0.14
Soluble fiber (g/100 g)	3.25 ± 0.03

**Table 2 plants-14-03350-t002:** Fatty acids (relative percentage) of *Atriplex halimus* leaves.

Fatty Acid	*A. halimus* Leaves %
C12:0 (lauric)	2.6 ± 0.6
C14:0 (myristic)	6.1 ± 1.8
C16:0 (palmitic)	46.9 ± 8.5
C18:0 (stearic)	1.7 ± 1.5
C18:1n9c (oleic)	2.1 ± 0.7
C18:2n6c (linoleic)	23.7 ± 3.5
C18:3n3 (α-linolenic)	16.8 ± 3.4
Total SFA	57.3 ± 7.7
Total MUFA	2.1 ± 0.7
Total PUFA	40.6 ± 7.0

**Table 3 plants-14-03350-t003:** Mineral determination of *Atriplex halimus* leaves.

Minerals	*A. halimus* Leaves Content
Macro-elements	
Na (g/100 g)	9.09 ± 0.06
Mg (g/100 g)	1.16 ± 0.04
K (g/100 g)	1.56 ± 0.01
Ca (g/100 g)	1.16 ± 0.04
Microelements	
Fe (mg/100 g)	142.0 ± 2.41
Zn (mg/100 g)	2.40 ± 0.16
Cu (mg/100 g)	0.46 ± 0.06
Mn (mg/100 g)	11.07 ± 0.23

**Table 4 plants-14-03350-t004:** Total polyphenolic content (TPC), DPPH, and FRAP antioxidant activity.

Samples	TPC (mg GAE/g DW)	DPPH IC_50_ (µg/mL)	DPPH (µmol TE/g DW)	FRAP (µmol TE/g DW)
Hydroalcoholic extract	30.12 ± 1.01	827 ± 24	22.71 ± 1.68	60.54 ± 0.54
Decoction	27.36 ± 0.82	1798 ± 80	7.07 ± 0.37	64.84 ± 4.08
Trolox		4.37 ± 0.31		

**Table 6 plants-14-03350-t006:** Bioaccessible total polyphenols in the salivary, gastric, intestinal phases, and total phenolic contents in the decoction of crude drug (CD) and in the decoction of solid residue from the intestinal phase (final solid residue).

	Salivary Phase (%)	Gastric Phase (%)	Intestinal Phase (%)
Total bioaccessibility	64.89 ± 0.54	69.13 ± 1.04	71.52 ± 0.46
	**Crude drug decoction** (mg GAE/100 g CD)		**Final solid residue decoction** (mg GAE/100 g CD)
Total phenolic content	633 ± 40		85 ± 6

**Table 7 plants-14-03350-t007:** Sample responses in TA98 and TA1535, with and without S9. Results are expressed in colony-forming units (CFU) per plate (mean ± SD).

Extracts	Concentration (mg/plate)	TA98 (CFU/Plate)		TA1535 (CFU/Plate)	
		Without S9	With S9	Without S9	With S9
UAE	0 (DMSO)	23.00 ± 1.41	37.33 ± 0.71	18.00 ± 1.41	16.33 ± 2.12
	0.5	31.67 ± 3.06	38.33 ± 5.51	10.67 ± 0.58	11.67 ± 3.21
	2.5	18.67 ± 2.08	30.33 ± 5.13	10.67 ± 1.53	10.33 ± 3.06
	5	30.33 ± 2.89	35.00 ± 3.00	16.00 ± 1.73	13.00 ± 6.56
Decoction	0 (DMSO)	21.00 ± 0.00	29.00 ± 4.24	18.00 ± 1.41	16.67 ± 2.12
	0.5	21.33 ± 1.53	25.33 ± 4.16	10.67 ± 0.58	11.67 ± 3.21
	2.5	29.00 ± 1.00	32.00 ± 3.61	10.67 ± 1.53	10.33 ± 3.06
	5	29.00 ± 1.00	34.67 ± 1.53	16.00 ± 1.73	13.00 ± 6.56
Experimental range *		12–32	18–38	7–19	8–18
Literature range **		20–50	20–50	5–20	5–20

* Experimental spontaneous reversion range has been extrapolated from previous experiments conducted in our laboratory. ** Range riported in literature [36,37].

## Data Availability

The original contributions presented in the study are included in the article; further inquiries can be directed to the corresponding author.

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
