# Peer review of "Investigating the Nutritional Properties, Chemical Composition (UPLC-HR-MS) and Safety (Ames Test) of *Atriplex halimus* L. Leaves and Their Potential Health Implications"

_plants, 2025, doi:10.3390/plants14213350_

Round 1
Reviewer 1 Report
Comments and Suggestions for Authors
The paper entitled “Investigating the chemical properties and safety of Atriplex halimus L. leaves and their health implications for Sahrawi refugees” is a scientific contribution in the fields of nutrition, chemistry and pharmacy of plants. However, the way of approaching of the manuscript concerning the population of Sahrawi es worrying.
From the title and throughout the entire text of the manuscript, references to the Sahrawi population seem inappropriate. In the Abstract (lines 21-23) the traditional use that the Sahrawi people have exercised of this Atriplex halimus shrub is mentioned, and this is part of the background and justification for studying the shrub, however, the reason why information was included (lines 52-54) is not explained, such as the fact that the Sahrawi people have lived as refugees... what is the relevance of this information in the scientific study? The same occurs in lines 69-69, 76, 90-91, and 112-113, where references are made to the Sahrawi people about suffering from diabetes or having nutritional deficiencies, as if they were characteristics that define or should be highlighted from said population for the scientific study.
Therefore, from my point of view, despite the technical-scientific value of the manuscript and its contribution to the nutritional knowledge of the Atriplex halimus shrub, the manuscript must be refocused in its entirety to take only as background the valuable traditional use that the Sahrawi people have had of the shrub and that provides base knowledge for the research. References that could be interpreted as stigmas or characteristics of a population must be completely eliminated.
Reviewer 2 Report
Comments and Suggestions for Authors
Dear authors,
I had the opportunity to review the manuscript No.: plant-3914119 with title:” Investigating the chemical properties and safety of Atriplex halimus L. leaves and their health implications for Sahrawi refugees”.
After reviewing the article and conducting my own research, I discovered that the topic addressed makes an important contribution by providing new information about the use of edible plants as a source of nutritional value that must be rigorously evaluated from a chemical and toxicological standpoint.
Thus, the authors studied the leaves of the shrub Atriplex halimus L., which are traditionally used in Sahrawi culture as food, spice, fodder, and medicine. They play an important role and are used for their nutritional and therapeutic benefits.
To demonstrate their chemical and toxicological safety, the authors conducted a study on the plant's properties, analyzing the chemical, mineral composition, and biological activity of the leaves using modern methods (HR-UPLC-DAD-MS, antioxidant tests, in vitro digestion, Ames test).
The results showed a high nutritional value with a significant content of proteins, fibers (especially insoluble), essential fatty acids (linoleic and α-linolenic), and minerals (Ca, K, Fe, Mn, Zn). These characteristics support metabolic health and can help the Sahrawi population overcome common deficiencies like anaemia and growth disorders. Rare polyphenols and flavonoids were also identified, such as quercetin, isorhamnetin, and myricetin. DPPH and FRAP tests revealed moderate antioxidant activity, as well as high polyphenol bioaccessibility during simulated in vitro digestion. Finally, the Ames test revealed that the extracts did not cause mutagenic or genotoxic effects.
All of this demonstrates the plant's potential as a functional food and health supplement, particularly in vulnerable communities like the Sahrawis.
As a result, the manuscript is well-structured, contains new and interesting scientific information, and is entirely original.
Following my analysis, I decided that the manuscript can be published in the format presented.

Reviewer 3 Report
Comments and Suggestions for Authors
In my opinion, the study has major weaknesses that significantly limit its scientific impact. The novelty is very limited, as most chemical and nutritional findings reproduce previously reported data, without the discovery of new bioactive compounds or meaningful pharmacological evidence. Biological validation is weak, as the antioxidant results are negligible, and the only toxicological assay performed (the Ames test) provides insufficient support for human health relevance. The phytochemical analysis is not robust, since the identification of flavonoids is based solely on LC-MS fragmentation without structural confirmation through isolation or NMR. Furthermore, the manuscript contains overstated claims, suggesting relevance for addressing anemia and diabetes without in vivo data or functional validation, making such conclusions speculative. Finally, although the ethnopharmacological context of the Sahrawi refugees is introduced, there is no real nutritional or clinical data linking the experimental findings to population health outcomes. For these reasons, I recommend rejection, as the manuscript lacks sufficient novelty, depth, and biological relevance.
Reviewer 4 Report
Comments and Suggestions for Authors
Good day.
Please see the attached for your attention.
Thank you

Round 2
Reviewer 1 Report
Comments and Suggestions for Authors
Please remove the phrase "use in favor of the Sahrawi population" from line 111.
Author Response
Thank you for your positive comments.
We remove the phrase "use in favor of the Sahrawi population" from line 111.
Reviewer 3 Report
Comments and Suggestions for Authors
After considering all the revisions made, I believe that the manuscript may now be suitable for consideration for publication in its current form.
Author Response
Thank you for your positive comments.
Reviewer 4 Report
Comments and Suggestions for Authors
The manuscript has been improved as required by the Reviewers. I then recommend acceptance in its current form
Author Response
Thank you for your positive comments.